# ScreenCoder: Advancing Visual-to-Code Generation for Front-End Automation via Modular Multimodal Agents

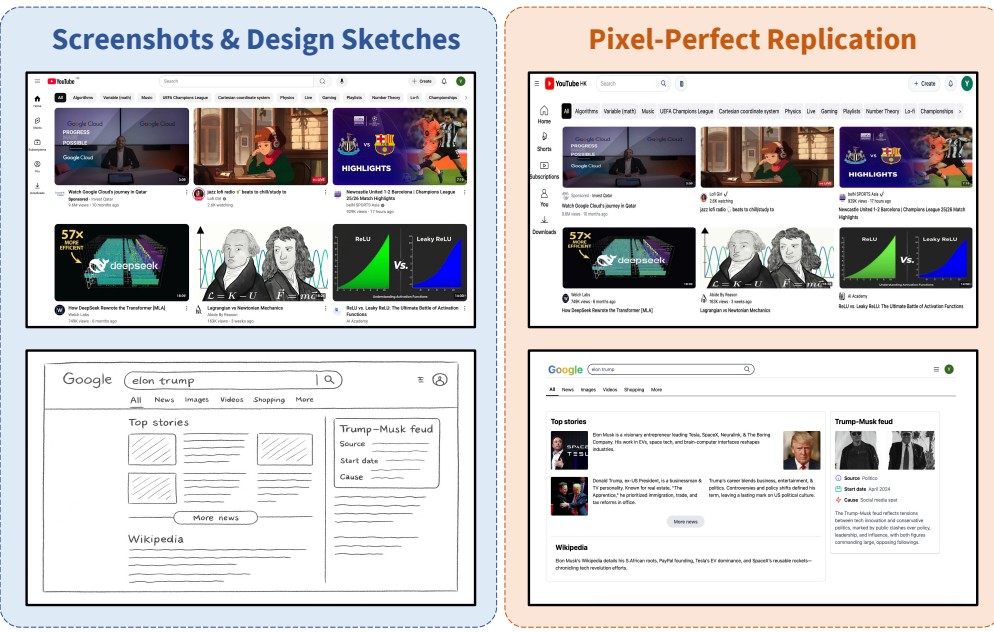

Figure 1: **ScreenCoder accurately transforms website screenshots and design sketches into pixel-perfect front-end code.** The figure showcases a variety of inputs on the left, including high-fidelity screenshots and a low-fidelity design sketch. The right column displays the corresponding webpages rendered from our model's generated code, demonstrating its high-fidelity replication capabilities.

## Abstract

Automating the transformation of user interface (UI) designs into front-end code holds significant promise for accelerating software development and democratizing design workflows. While multimodal large language models (MLLMs) can translate images to code, they often fail on complex UIs, struggling to unify visual perception, layout planning, and code synthesis within a single monolithic model, which leads to frequent perception and planning errors. To address this, we propose ScreenCoder, a modular multi-agent framework that decomposes the task into three interpretable stages: grounding, planning, and generation. By assigning these distinct responsibilities to specialized agents, our framework achieves significantly higher robustness and fidelity than end-to-end approaches. Furthermore, ScreenCoder serves as a scalable data engine, enabling us to generate high-quality image-code pairs. We use this data to fine-tune open-source MLLM via a dual-stage pipeline of supervised fine-tuning and reinforcement learning, demonstrating substantial gains in its UI generation capabilities. Extensive experiments demonstrate that our approach achieves state-of-the-art performance in layout accuracy, structural coherence, and code correctness.

# 1 INTRODUCTION

Automating front-end engineering is a critical step toward efficient software development, and recent large language models (LLMs) have advanced the generation of code directly from text instructions (Qwen, 2025; Bolt, 2025). However, this text-based approach faces significant limitations. Generating detailed UIs requires verbose prompts to capture structure and styling, struggles to specify fine-grained visual design like spacing or alignment, and fundamentally deviates from practical design workflows that begin with visual sketches, not paragraphs of text. Relying solely on textual input is therefore sub-optimal for real-world deployment and often fails to capture the full visual intent.

To bridge this gap, multimodal large language models (MLLMs) offer the promise of directly interpreting UI design images and translating them into code (Yang et al., 2023). While conceptually appealing, our analysis reveals that current MLLMs struggle with this task as it requires a unified set of capabilities, visual understanding, structural layout planning, and domain-specific code synthesis, that they are not holistically designed for. Empirically, this leads to two recurring failure modes: (1) perception errors, where components are missed or misclassified, and (2) planning errors, where components are placed incorrectly or violate layout constraints.

To address these limitations, we propose **ScreenCoder**, a modular multi-agent framework that decomposes the UI-to-code task into three interpretable stages: grounding, planning, and generation. The grounding agent leverages a multimodal large language model to localize and semantically label key UI regions. The planning agent then constructs a hierarchical layout tree using domain knowledge of web layout systems. Finally, the generation agent produces HTML and CSS code via adaptive prompt-based synthesis, incorporating both layout context and optional user instructions to support interactive design. This decomposition introduces architectural modularity, enabling more robust component recognition, layout planning, and structured code generation than end-to-end black-box methods. Experiments show that our framework achieves state-of-the-art performance in layout fidelity, structural coherence, and generation quality.

Beyond inference, our framework acts as a **scalable data engine**. This is crucial because training on raw web data is often infeasible, as its length and noise from dependencies and scripts destabilize training and prevent models from learning the core visual-to-code mapping (Si et al., 2025). To address the challenge, we leverage ScreenCoder to create **Screen-10K**, a new large-scale training dataset of 10,000 high-quality image-code pairs, curated by filtering an initial crawl of 50,000 webpages. We use Screen-10K to significantly enhance open-source MLLM via a dual-stage supervised fine-tuning and reinforcement learning pipeline. Furthermore, to facilitate a more rigorous evaluation of modern models, we introduce **ScreenBench**, a challenging new benchmark composed of 1,000 high-quality, up-to-date websites that reflect contemporary web design. Our framework thus provides a practical path for both scalable dataset creation and robust model alignment. To sum up, our contributions are as follows:

- We conduct a systematic investigation into the limitations of existing MLLMs on UI-to-code tasks and propose a novel modular multi-agent framework that decomposes the complex UI-to-code generation task into three interpretable stages: grounding, planning, and generation, significantly outperforming existing end-to-end multimodal models in layout fidelity and structural coherence.

- Leveraging our framework as a scalable data engine, we introduce Screen-10K, a new large-scale dataset containing 10,000 high-quality image-code pairs, which addresses a critical bottleneck in the field by providing a substantial resource for training more capable UI-to-code generation models.

- To facilitate more rigorous and relevant evaluation, we present ScreenBench, a new challenging benchmark of 1,000 diverse and contemporary web designs. ScreenBench provides a more accurate measure of model performance on real-world tasks compared to existing benchmarks.

## 2 RELATED WORK

### 2.1 MULTIMODAL LARGE LANGUAGE MODELS

Multimodal Large Language Models (MLLMs) integrate vision and text to enable joint reasoning. Early models like VisualGPT (Chen et al., 2022) and Frozen (Tsimpoukelli et al., 2021) pioneered this by using pre-trained LLMs to decode visual features. Architectural innovations, such as Flamingo's (Alayrac et al., 2022) gated cross-attention and BLIP-2's (Li et al., 2023) Q-Former, further improved vision-language alignment. Modern systems like Gemini 2.5 (Google, 2024) and GPT-4o (OpenAI, 2024) have dramatically scaled these capabilities, excelling at complex multimodal tasks and enabling applications like website generation from images (Zhu et al., 2023). However, despite their impressive general-purpose abilities, these models struggle with domain-specific structured generation, such as UI-to-code synthesis. This limitation stems from a lack of inductive biases for spatial layout and hierarchical planning, as well as a monolithic architecture that hinders the injection of task-specific knowledge.

### 2.2 VISUAL-TO-CODE GENERATION

Early visual-to-code methods used CNNs and LSTMs to translate UI screenshots into domain-specific languages (DSLs) (Beltramelli, 2018), which offered limited real-world applicability (Xu et al., 2021). Subsequent research shifted towards generating general-purpose HTML/CSS (Chen et al., 2018) and improving component recognition, layout understanding (Cizotto et al., 2023), interaction-awareness (Xiao et al., 2024), and multi-page generation (Wan et al., 2024). Alternative approaches have included OCR-based techniques (Nguyen & Csallner, 2015) and object detection for screen parsing (Wu et al., 2021). More recently, divide-and-conquer methods (Wan et al., 2025; Wu et al., 2025; Gui et al., 2025b;a) have emerged, but often depend on heuristic-based segmentation. Despite these advances, prior works are often brittle, rely on synthetic data, and lack the interpretability needed to jointly model complex visual semantics, layouts, and coding patterns. In contrast, our approach introduces a modular multi-agent framework that decomposes the task into interpretable sub-tasks: grounding, planning, and generation. This allows for explicit reasoning and the integration of domain-specific priors. Our system also functions as a scalable data engine to train future MLLMs, addressing the scarcity of high-quality, realistic image-code datasets.

## 3 MOTIVATION: WHY MLLMS FAIL AT UI-TO-CODE GENERATION

To motivate our approach, we analyze why state-of-the-art multimodal large language models (MLLMs) struggle with the direct, end-to-end generation of code from UI screenshots. While models like GPT-4o exhibit powerful general visual reasoning, our analysis of their outputs on real-world webpages reveals two primary failure modes, as illustrated in Figure 2: perception failures and planning failures.

First, **perception failures** stem from the model's inability to accurately interpret visual elements. This materializes in two ways: (1) **element omission**, where smaller or less salient components like icons and secondary text are completely missed, and (2) **element distortion**, where an element is recognized but its attributes (e.g., text content, color) are incorrect, or its type is misidentified (e.g., an input field rendered as static text). Second, **planning failures** involve the inability to organize perceived elements into a coherent spatial and hierarchical structure. Even if all components are identified, they are often assembled incorrectly, leading to (1) **element misarrangement**, where components are placed in the wrong positions, and (2) **hierarchical incoherence**, where the model produces a "flat" layout that fails to infer the nested, DOM-like structure essential for modern, responsive web design. This highlights a lack of inductive bias for fundamental front-end layout conventions.

Our analysis indicates these failures arise not from an intrinsic inability to perform any single subtask, but from the burden of a monolithic, end-to-end approach that overloads a single model with granular perception, complex spatial planning, and structured code synthesis simultaneously. This task also demands deep domain knowledge of front-end development, such as layout systems and component hierarchies, which general-purpose MLLMs lack. Inspired by agentic workflows that decompose complex problems, we hypothesize that separating these distinct challenges will yield

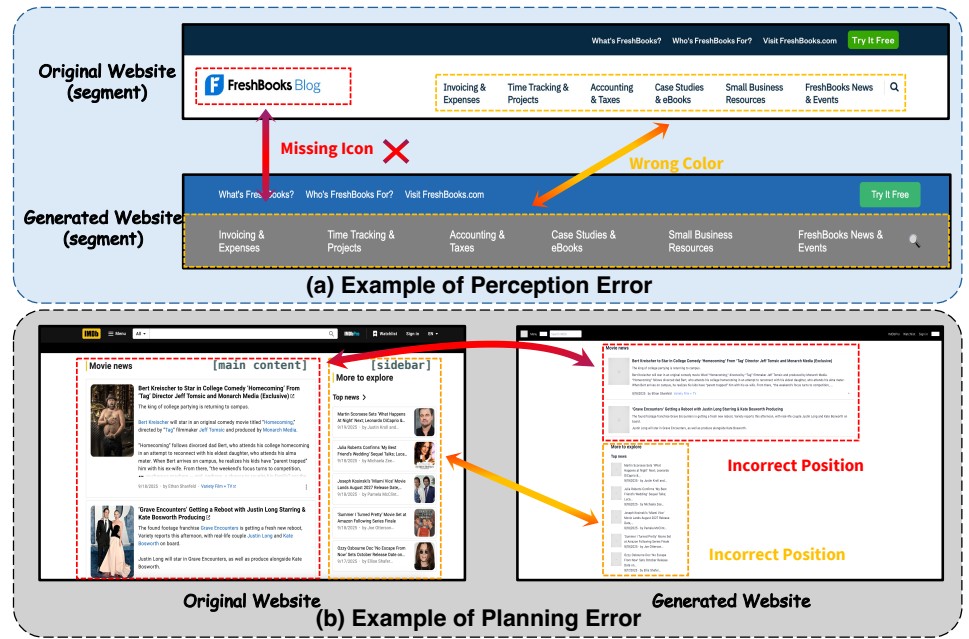

Figure 2: **Analysis of Common MLLM Failure Modes in UI-to-Code Generation.** We identify two primary error categories: (a) Perception Errors, where the model fails to accurately interpret visual details, leading to missing icons or incorrect colors, and (b) Planning Errors, where the model fails to correctly reason about the spatial layout, resulting in elements being placed in the wrong positions.

more robust results. This insight directly motivates our modular, multi-agent framework. By explicitly decoupling the task into **grounding** (to address perception failures), **planning** (to address structural failures), and **generation** (to focus on code synthesis), we enable each agent to specialize. Crucially, this modularity allows us to inject specific domain knowledge at each stage, such as established layout conventions in the planning agent, thereby mitigating the failure modes inherent in a single, end-to-end process.

## 4 METHOD

We propose a modular, multi-agent framework for UI-to-code generation that decomposes the complex task into three sequential agents: grounding, planning, and generation. This design is explicitly motivated by the failure modes identified in Section 2; each agent is specialized to address a distinct sub-problem, allowing the system to leverage both visual understanding and structured reasoning in a coordinated manner. The grounding agent targets *perception errors* by accurately identifying UI components. The planning agent tackles *planning errors* by constructing a coherent layout hierarchy. Finally, the generation agent translates this structured plan into high-fidelity code. Our overall framework is shown in Figure 3.

### 4.1 GROUNDING AGENT: OVERCOMING PERCEPTION ERRORS

The grounding agent serves as the perceptual front-end of our framework, tasked with detecting and semantically labeling major structural components to overcome the *perception errors* (element omission and distortion) common in end-to-end models. This design choice, assigning explicit labels like `sidebar`, `header`, and `navigation`, is crucial for enabling interactive, language-driven design, as it allows users and downstream agents to reference and manipulate specific components via natural language (*e.g.*, "resize the sidebar").

To accomplish this, the agent employs a Multimodal Large Language Model (MLLM) queried with prompts such as "Where is the sidebar?" or "Locate the header area." The MLLM returns a set of

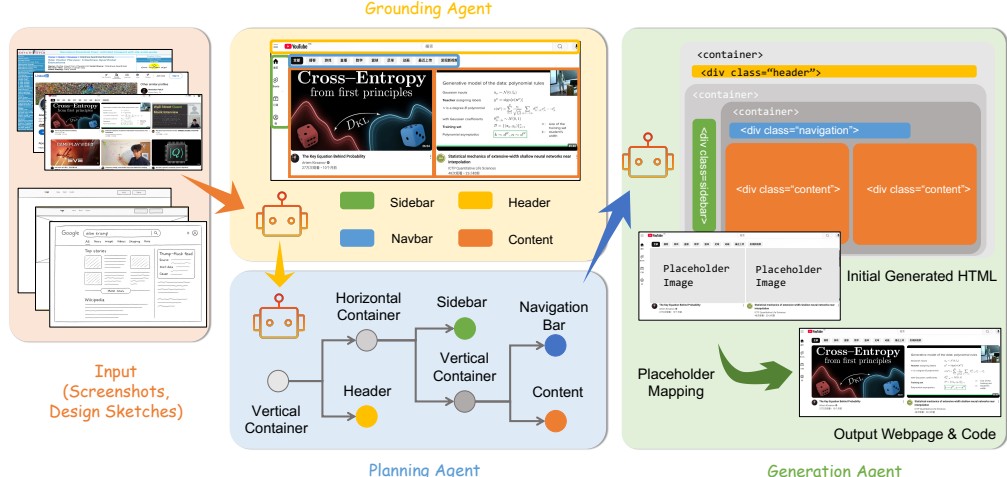

Figure 3: **Overview of ScreenCoder.** Given UI screenshots or design sketches as input, the Grounding Agent first detects and labels key components (*e.g.*, header, navbar, sidebar, content). The Planning Agent organizes these components into a hierarchical layout using front-end engineering priors. The Generation Agent synthesizes initial HTML code with placeholders, followed by content mapping to produce the final webpage and code.

grounded regions as bounding boxes and their associated labels:

$$\mathcal{B} = \{(b_i, l_i) \mid l_i \in \mathcal{L}\}_{i=1}^{N}, \text{ where } \mathcal{L} = \{\texttt{sidebar}, \texttt{header}, \texttt{navigation}\}. \quad (1)$$

Here, each $b_i = (x_i, y_i, w_i, h_i)$ is a bounding box in pixel coordinates. Unlike traditional object detection, this MLLM-based approach allows grounding to be flexibly guided by text, making the system extensible to new UI concepts.

To ensure robustness, the agent performs several post-processing operations: (1)Deduplication and Conflict Resolution, using class-specific non-maximum suppression (NMS) to filter multiple detections for the same label and retain the most confident one; (2)Fallback Recovery, invoking a heuristic based on spatial priors if a key component is missed (*e.g.*, a wide, short box at the top is likely a header); and (3)Main Content Inference, which robustly defines the primary content area by inferring it as the largest rectangular area not overlapping any detected component. The final output is a layout dictionary which provides the semantic and spatial foundation for the next stage. Unlike traditional object detectors which require costly retraining, our MLLM-based approach is inherently extensible. The system can be adapted to recognize new UI concepts simply by expanding the textual label set $\mathcal{L}$, offering a flexible path for future domain adaptations.

## 4.2 PLANNING AGENT: CORRECTING PLANNING ERRORS

The Planning Agent mitigates common MLLM failures in spatial reasoning, such as component misarrangement and hierarchical incoherence. Instead of a generative approach, it uses a novel, deterministic Visual-to-Structural Tree Mapping algorithm1 to programmatically translate unstructured visual information into a well-formed layout. The algorithm embeds key domain knowledge from modern web development by converting the flat 2D canvas of bounding boxes into a DOM-like tree, the standard hierarchical data structure for web pages. This deliberately trades the unconstrained flexibility of generative models for structural integrity and predictability, ensuring the generated code faithfully mirrors the source screenshot's layout.

First, our algorithm establishes the global layout by creating tree nodes for primary components (e.g., header, sidebar), converting absolute pixel coordinates into responsive percentage-based values, and using absolute positioning to preserve the macro-structure. It then recursively handles internal component layouts by injecting domain knowledge, designating parent regions with children as CSS Grid containers to arrange nested elements with high fidelity. This process yields an interpretable layout tree that serves as a blueprint, effectively decoupling the abstract planning of

---

**Algorithm 1** Visual-to-Structural Tree Mapping

---

1: **Input:** Layout dictionary $L = \{l \mapsto b_l\}$, Image dimensions $W, H$
2: **Output:** Layout Tree $\mathcal{T}$
3: $\mathcal{T} \leftarrow$ createNode('root', style={'position': 'relative', 'width': '100%', ...})
4: **for all** label $l$, box $b_l$ in $L$ **do**
5: $\quad N_l \leftarrow$ createNode($l$)
6: $\quad (x_\%, y_\%, w_\%, h_\%) \leftarrow (b_l.x/W, b_l.y/H, b_l.width/W, b_l.height/H) \times 100$
7: $\quad N_l$.style $\leftarrow$ {'position': 'absolute', 'left': $x_\%$, 'top': $y_\%$, ...}
8: $\quad$ **if** $l$ contains subdivisions **then** $N_l$.is_grid_container $\leftarrow$ **true**
9: $\quad$ **end if**
10: $\quad$ AddChild($\mathcal{T}, N_l$)
11: **end for**
12: **return** $\mathcal{T}$

---

the UI structure from the concrete task of code generation, embodying the principle of separation of concerns.

### 4.3 GENERATION AGENT: HIGH-FIDELITY CODE SYNTHESIS

The generation agent translates the hierarchical layout tree $\mathcal{T}$ into executable HTML and CSS. It traverses the tree and, for each node, uses a large language model to generate code based on an *adaptive prompt*. This prompt combines the component's semantic label, its structural context from the tree, and optional user instructions. This approach provides the LLM with rich context, guiding it to produce code that is not only visually correct but also structurally sound and responsive to user intent. The generated code snippets for each component are then assembled according to the tree structure, preserving the hierarchy and layout defined by the planning agent. This closes the loop from visual perception to structured, interactive code synthesis.

### 4.4 PLACEHOLDER MAPPING

To restore visual fidelity, we introduce a final placeholder mapping stage that replaces generic image placeholders with their original visual assets. The process begins by using a UI Element Detection (UIED)(Xie et al., 2020) model to extract all visual elements (e.g., icons, images) from the source screenshot. These elements are then partitioned according to the semantic regions defined by the Planning Agent (e.g., header, sidebar).

Within each region, we solve an optimal assignment problem to match the detected elements to the placeholders. We construct a cost matrix based on the negative Complete IoU (CIoU) between the placeholder boxes and the original element boxes, after applying a localized affine transformation to correct for minor rendering discrepancies. This bipartite matching problem is solved using the Hungarian Algorithm to find the optimal one-to-one mapping. Finally, the matched image patches are cropped from the original screenshot and inserted into the generated code, restoring the full visual content of the UI.

## 5 ENHANCING MLLMS WITH SCALABLE DATA GENERATION AND DUAL-STAGE POST-TRAINING

Beyond its inference capabilities, our framework serves a crucial role as a scalable data engine, addressing a fundamental challenge in training visual-to-code models. Directly training on raw, crawled web data is often infeasible (Si et al., 2025). Real-world code is typically long and noisy, replete with complex dependencies, external links, and irrelevant scripts that make training unstable and hinder the model's ability to learn the core mapping from visual structure to clean, self-contained code.

To overcome this, we leverage our engine to generate Screen-10K, a large-scale dataset of 10,000 high-quality image-code pairs. This dataset was curated by initially crawling 50,000 webpages and applying a rigorous, automated filtering process to retain only the most valuable, well-structured ex-

| Model | ScreenBench | | | | | Design2Code | | | | |
|-------|-------|------|----------|-------|------|-------|------|----------|-------|------|
| | Block | Text | Position | Color | CLIP | Block | Text | Position | Color | CLIP |
| GPT-4o | 0.745 | 0.835 | 0.725 | 0.702 | 0.775 | 0.845 | 0.962 | 0.903 | 0.881 | 0.917 |
| GPT-4V | 0.721 | 0.815 | 0.701 | 0.682 | 0.758 | 0.831 | 0.955 | 0.895 | 0.872 | 0.905 |
| Gemini-2.5-Pro | 0.741 | 0.825 | 0.752 | 0.695 | 0.788 | 0.841 | 0.969 | 0.901 | 0.879 | 0.908 |
| LLaVA 1.6-7B | 0.635 | 0.830 | 0.544 | 0.592 | 0.727 | 0.736 | 0.910 | 0.729 | 0.816 | 0.802 |
| DeepSeek-VL-7B | 0.680 | 0.773 | 0.570 | 0.614 | 0.732 | 0.718 | 0.824 | 0.702 | 0.720 | 0.843 |
| Qwen2.5-VL | 0.723 | 0.828 | 0.613 | 0.632 | 0.762 | 0.822 | 0.951 | 0.815 | 0.831 | 0.893 |
| Seed1.5-VL | 0.727 | 0.852 | 0.742 | 0.729 | 0.783 | 0.829 | 0.968 | 0.915 | 0.897 | 0.911 |
| DCGen | 0.731 | 0.831 | 0.713 | 0.699 | 0.767 | 0.836 | 0.958 | 0.885 | 0.865 | 0.901 |
| Websight-8B | 0.678 | 0.768 | 0.554 | 0.606 | 0.748 | 0.755 | 0.903 | 0.767 | 0.785 | 0.859 |
| ScreenCoder (Agentic) | **0.768** | **0.857** | **0.755** | **0.734** | **0.812** | **0.865** | **0.975** | **0.925** | **0.908** | **0.922** |
| ScreenCoder (Finetuned) | 0.758 | 0.841 | 0.742 | 0.718 | 0.791 | 0.849 | 0.968 | 0.913 | 0.886 | 0.915 |

Table 1: Automatic evaluation results on the ScreenBench and Design2Code benchmarks. For each metric, the best result is in **bold** and the second best is underlined.

amples. This clean dataset provides the stable foundation necessary for our dual-stage post-training pipeline. First, we perform supervised fine-tuning (SFT) on a 9,000-pair subset to align the model's visual understanding with correct code syntax, establishing a strong baseline. The remaining 1,000 pairs are then used in a subsequent reinforcement learning (RL) stage to further optimize for high visual fidelity.

The reinforcement learning stage is based on Group Relative Policy Optimization (GRPO) (Shao et al., 2024). We optimize the policy $\pi_\theta$ to maximize the expected reward over our RL dataset:

$$\max_\theta \ \mathbb{E}_{(x,y)\sim\pi_\theta}\left[\mathcal{R}(x,y)\right], \tag{2}$$

where $x$ is the input image and $y$ is the generated code.

To directly optimize for visual fidelity, our reward function $\mathcal{R}(x,y)$ is based on the pixel-level similarity between the original screenshot and the webpage rendered from the generated code. For each output $y$, we first render it to produce an image, $\text{Render}(y)$. The reward is then defined as the negative Mean Squared Error (MSE) between the original image $x$ and the rendered output. Since RL seeks to maximize reward, using a negative error term incentivizes the policy to minimize the pixel-wise difference:

$$\mathcal{R}(x,y) = -\text{MSE}(x, \text{Render}(y)) \tag{3}$$

where the MSE between two images $I_1$ and $I_2$ of height $H$ and width $W$ is calculated as:

$$\text{MSE}(I_1, I_2) = \frac{1}{H \times W} \sum_{i=1}^{H} \sum_{j=1}^{W} \|I_1(i,j) - I_2(i,j)\|_2^2 \tag{4}$$

This reward function provides a strong, holistic signal that guides the model toward generating code that achieves a pixel-perfect visual replication of the original design.

# 6 EXPERIMENTS

We evaluate our framework from two complementary perspectives: (1) the visual fidelity and semantic consistency of the generated webpages, and (2) its effectiveness as a scalable data engine for fine-tuning multimodal large language models (MLLMs). We assess the quality of the generated code by comparing the rendered output against ground-truth screenshots using the metrics and benchmarks detailed below.

## 6.1 EXPERIMENTAL SETUP

**Datasets.** To rigorously evaluate modern UI-to-code models, we introduce **ScreenBench**, a new benchmark of 1,000 high-quality image-code pairs. This benchmark addresses the limitations of existing datasets like Design2Code (Si et al., 2025), which, with its 484 samples from older websites, primarily tests for textual content reproduction over structural complexity. In contrast, Screen-

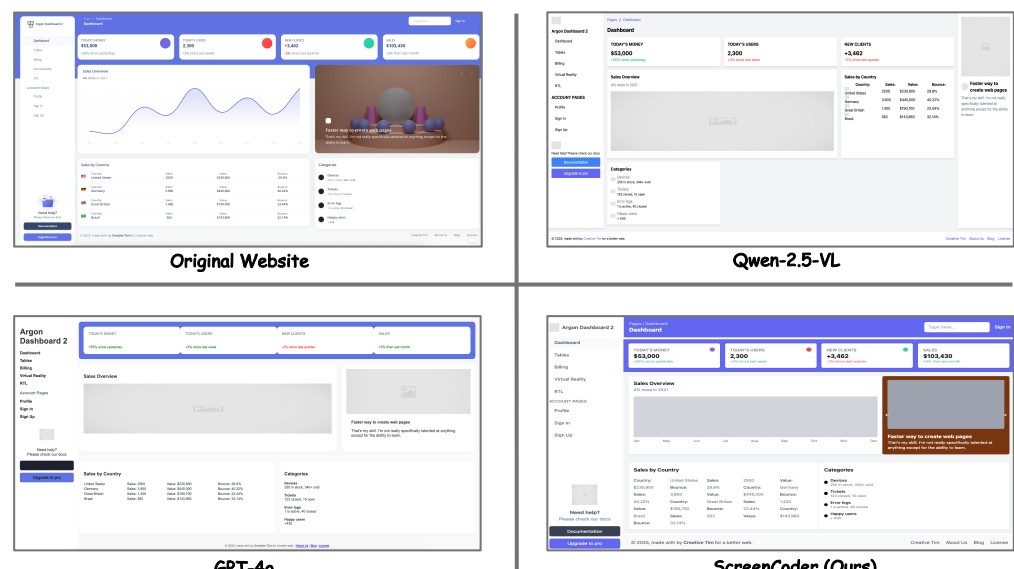

Figure 4: **Qualitative comparison of UI-to-code generation.** While leading MLLMs fail to accurately replicate the target website's layout, styling, and component structure, our method, Screen-Coder, produces a high-fidelity result that closely matches the original design in both appearance and organization.

Bench is substantially larger and sourced from contemporary web applications, featuring the complex, nested layouts (e.g., CSS Grid/Flexbox) that define modern web design. We also adopt Design2Code (Si et al., 2025) for evaluation.

**Evaluation Metrics.** We follow the methodology of Design2Code (Si et al., 2025) and evaluate visual similarity using both high-level and low-level metrics. For high-level assessment, we compute the CLIP similarity (Radford et al., 2021) between the rendered output and the reference screenshot. For low-level evaluation, we extract OCR-based visual blocks from both images, align them using text similarity, and then measure four key aspects based on these matched elements: block reproduction accuracy, textual consistency, spatial alignment, and color similarity.

**Baselines.** We benchmark our approach against a comprehensive suite of state-of-the-art models. This includes leading proprietary MLLMs (GPT-4o (OpenAI, 2024), GPT-4V (OpenAI, 2023), and Gemini-2.5-Pro (Google, 2024)); a range of open-source MLLMs (LLaVA 1.6-7B (Liu et al., 2023), DeepSeek-VL-7B (Lu et al., 2024), Qwen2.5-VL (Bai et al., 2025), and Seed1.5-VL (Guo et al., 2025)); and specialized UI-to-code methods (DCGen (Wan et al., 2025) and Websight-8B (Laurençon et al., 2024)). Our method is implemented on the open-source Qwen2.5-VL-32B model. We show other implementation details in Appendix C.1 and C.2.

## 6.2 MAIN RESULTS

As shown in Table 1, our ScreenCoder framework, in both its agentic and fine-tuned variants, consistently surpasses all open-source baselines. The primary agentic model achieves state-of-the-art performance, outperforming even top proprietary systems on the challenging ScreenBench with a Block score of 0.755. Furthermore, our fine-tuned model secures the second-best results on many metrics, and achieve comparable performance with close-source models. These results validate our framework's dual utility as both a high-performing inference system and an effective data engine for significantly enhancing open-source MLLMs. Besides automatic evaluation, we also conduct human expert evaluation, whose setting and result are shown in Appedix A.

| Training Stage | ScreenBench | | | | | Design2Code | | | | |
|---|---|---|---|---|---|---|---|---|---|---|
| | Block | Text | Position | Color | CLIP | Block | Text | Position | Color | CLIP |
| Base Model (Qwen2.5-VL) | 0.723 | 0.828 | 0.613 | 0.632 | 0.762 | 0.822 | 0.951 | 0.815 | 0.831 | 0.893 |
| + SFT | 0.741 | 0.835 | 0.705 | 0.681 | 0.784 | 0.842 | 0.963 | 0.890 | 0.870 | 0.908 |
| + RL (Final) | **0.758** | **0.841** | **0.742** | **0.718** | **0.791** | **0.849** | **0.968** | **0.913** | **0.886** | **0.915** |

Table 2: Effect of different dual-stage post-training stage on the ScreenBench and Design2Code benchmarks.

### 6.3 QUALITATIVE ANALYSIS

Figure 4 provides a qualitative comparison that highlights the practical advantages of our method. Leading end-to-end MLLMs, such as Qwen-2.5-VL and GPT-4o, demonstrate significant perception and planning failures. They struggle to replicate the target design, resulting in distorted layouts, incorrect component arrangements, and a general loss of styling information. In stark contrast, ScreenCoder produces a high-fidelity result that closely mirrors the original website's appearance and organization. More qualitative results are shown in Figure 1 and Appendix.

### 6.4 ABLATION STUDY

To isolate the contributions of our dual-stage training, we conducted an ablation study (Table 2). Starting with the base Qwen2.5-VL model, the application of Supervised Fine-Tuning (SFT) yielded significant improvements across all metrics, especially in spatial layout awareness ('Position'). The subsequent Reinforcement Learning (RL) stage further refined the model's capabilities, providing incremental but crucial boosts to achieve our final, top-performing results. This analysis confirms that SFT builds a strong foundational understanding, while RL fine-tunes the model's precision, validating our cumulative training strategy.

## 7 DISCUSSION

**Interactive Design and Human-in-the-Loop Feedback.** One key strength of our modular pipeline is its potential to support interactive design iteration. Since each stage, grounding, planning, and generation, is disentangled, user feedback can be incorporated at different abstraction levels. For instance, designers can manually adjust the layout tree or re-prompt specific components without restarting the entire process. Future work may further enhance this interactivity by integrating real-time preview, editable intermediate representations, and dialogue-based refinement.

**Scalability and the Cost-Quality Trade-Off.** Our modular framework offers a flexible trade-off between computational cost and output quality via test-time scaling. Users can select smaller, faster models for quick, low-fidelity drafts or larger, more powerful models for high-fidelity, production-ready code. While a high-quality generation takes tens of seconds, this is an acceptable trade-off for its intended use as a workflow accelerator where initial code quality is the priority. This versatility allows the system to be adapted for different stages of the development process. Future work can further enhance this flexibility. We envision a system where developers can interactively refine the output, dedicating more compute only to the specific components that require changes. This moves beyond a one-off generation and towards a more dynamic, resource-efficient, human-in-the-loop partnership.

## 8 CONCLUSION

We present ScreenCoder, a modular multi-agent framework for UI-to-code generation that addresses key limitations of end-to-end models, and also functions as a scalable data engine to improve MLLMs via dual-stage post-training with supervised fine-tuning and reinforcement learning. Experiments demonstrate state-of-the-art performance in visual fidelity and code correctness, offering a practical solution for front-end automation and a foundation for future research in multimodal program synthesis.

ETHICAL STATEMENTS

This research was conducted in full compliance with the ICLR Code of Ethics. All aspects of our work, from data collection to model development, adhere to ethical standards of privacy, consent, and responsible AI. To the best of our knowledge, this study does not introduce any new ethical risks.

REPRODUCIBILITY STATEMENT

To ensure the reproducibility of our work, we have made several resources available. First, the complete implementation code is provided in the supplementary material. Second, our visual mapping algorithm is formally described in Algorithm 1. Third, the prompt templates used for each agent are detailed in Appendix C.1. Finally, a comprehensive breakdown of the training procedure is included in Appendix C.2.

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

APPENDIX

# A  HUMAN EVALUATION

To validate our automatic metrics, we conducted a human evaluation study assessing two critical dimensions: the **perceived visual fidelity** of the generated webpages and the **practical utility** of ScreenCoder in a real-world development workflow.

## A.1  PAIRWISE COMPARISON OF VISUAL FIDELITY

**Setup.**  We recruited six Ph.D. students with web development experience to serve as expert annotators. Following the standard methodology for evaluating generative models, we performed a pairwise comparison study. In each trial, annotators were shown an original UI screenshot alongside two rendered webpages—one generated by **ScreenCoder (Agentic)** and one by a strong baseline (**GPT-4o**). They were asked to vote for the page that was more visually similar to the original ("A is better," "B is better," or "Tie"). To ensure reliable judgments, each comparison was evaluated by three annotators, and a winner was declared based on a majority vote ($\geq 2$).

**Results.**  The results, summarized in Table 3, reveal a strong human preference for our method. Annotators judged ScreenCoder's output as superior to GPT-4o's in **65%** of cases, while finding it inferior in only **11%**. This outcome confirms that the improvements captured by our automatic metrics translate into a noticeably better perceptual experience, suggesting that our modular approach produces layouts that are more structurally coherent and visually faithful.

Table 3: Pairwise comparison of visual fidelity between ScreenCoder and GPT-4o. Results show the percentage of times each outcome was chosen by human annotators.

| Outcome | Win Rate (%) |
|---|---|
| ScreenCoder Wins | **65%** |
| GPT-4o Wins (Baseline) | 11% |
| Tie | 24% |

## A.2  WORKFLOW USEFULNESS STUDY

**Setup.**  To measure ScreenCoder's impact on developer productivity, we designed a task-based study. Six participants with front-end experience were divided into an **Experimental Group (n=3)**, who used a tool powered by ScreenCoder, and a **Control Group (n=3)**, who used their preferred method (all chose GPT-4o). Each participant was tasked with converting two UI screenshots (one simple, one complex) into functional webpages, with a 20-minute time limit for each task. We measured both the completion time and the quality of the final output, which was rated for UI similarity on a 5-point Likert scale by two independent expert judges.

**Results.**  As detailed in Table 4, ScreenCoder provides a significant boost to development efficiency. On average, the ScreenCoder group completed tasks **2.2 times faster** than the control group (8.5 minutes vs. 18.7 minutes). Notably, all participants using ScreenCoder finished comfortably within the time limit, whereas two tasks in the control group timed out. Furthermore, the quality of the output was substantially higher for the ScreenCoder group, which achieved an average similarity score of **4.6/5.0**, compared to **3.4/5.0** for the control group (with a strong inter-judge Pearson correlation of 0.78). These findings demonstrate that ScreenCoder acts as a powerful accelerator in a human-in-the-loop process, enabling developers to build UIs faster and with greater accuracy.

# B  ADDITIONAL QUALITATIVE COMPARISONS

To further demonstrate the effectiveness of our framework, this appendix provides an extended set of qualitative results. The following figures showcase side-by-side comparisons between the webpages generated by our method, ScreenCoder, and those produced by leading baseline models.

Table 4: Results of the workflow usefulness study. We compare the performance of developers using ScreenCoder against a control group using GPT-4o.

| Metric | ScreenCoder (Experimental) | GPT-4o (Control) |
|---|---|---|
| Avg. Completion Time (min) | **8.5** | 18.7 |
| Avg. UI Similarity (out of 5.0) | **4.6** | 3.4 |
| Tasks Timed Out (out of 6) | **0** | 2 |

These examples were selected to cover a diverse range of website styles and layout complexities. They serve to highlight the common failure modes of existing end-to-end models—such as incorrect spatial arrangements, missing UI elements, and distorted styling—and illustrate how ScreenCoder's modular, agentic approach successfully overcomes these challenges. As shown in the comparisons, our method consistently produces webpages with significantly higher visual fidelity and structural coherence, more closely matching the original design intent of the source screenshots.

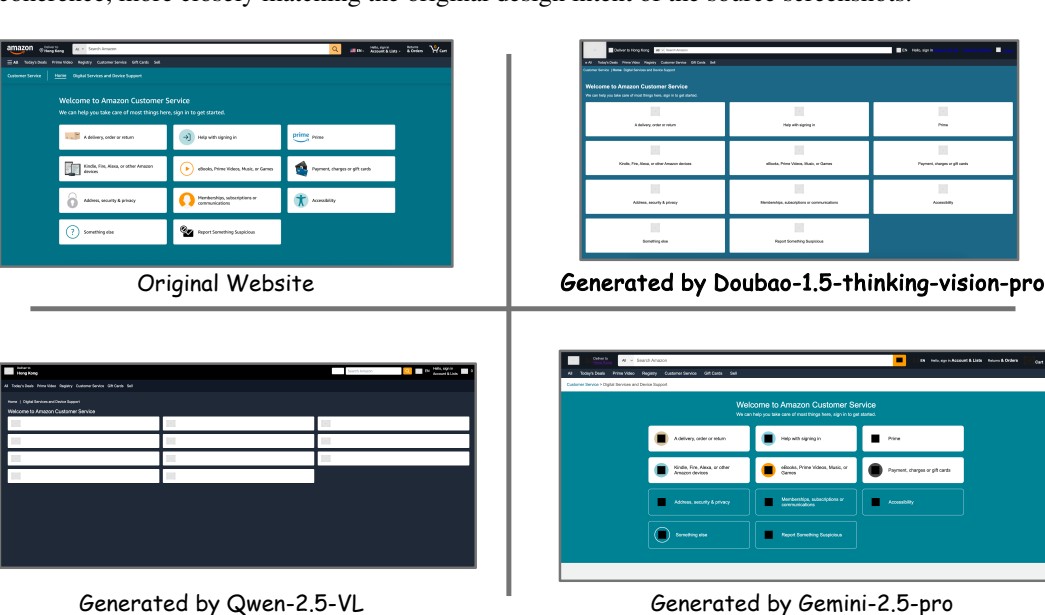

Figure 5: Qualitative comparison between our proposed method and various baselines.

## C  IMPLEMENTATION DETAIL

### C.1  PROMPT TEMPLATE

The prompt templates are shown in Figure 13 and 14.

### C.2  TRAINING DETAILS

Our training and experiments were conducted using the Llama Factory codebase on a high-performance cluster of 16 NVIDIA A100 GPUs. We adopted a two-stage training pipeline. For the SFT stage, we fine-tuned the base model on 9,000 samples from our Screen-10K dataset. We adopt the LLaMa Factory (Zheng et al., 2024) as the code base. The model was optimized using the AdamW optimizer with a cosine learning rate schedule, a peak learning rate of $2 \times 10^{-5}$, a weight decay of 0.01, and a warmup ratio of 10%. In the subsequent RL stage, we used the remaining 1,000 samples from Screen-10K. We employed the Group Relative Policy Optimization (GRPO) algorithm, continuing from the SFT checkpoint. The reward function was based on the negative Mean Squared Error (MSE) between the rendered output and the ground-truth image, directly optimizing for visual fidelity.

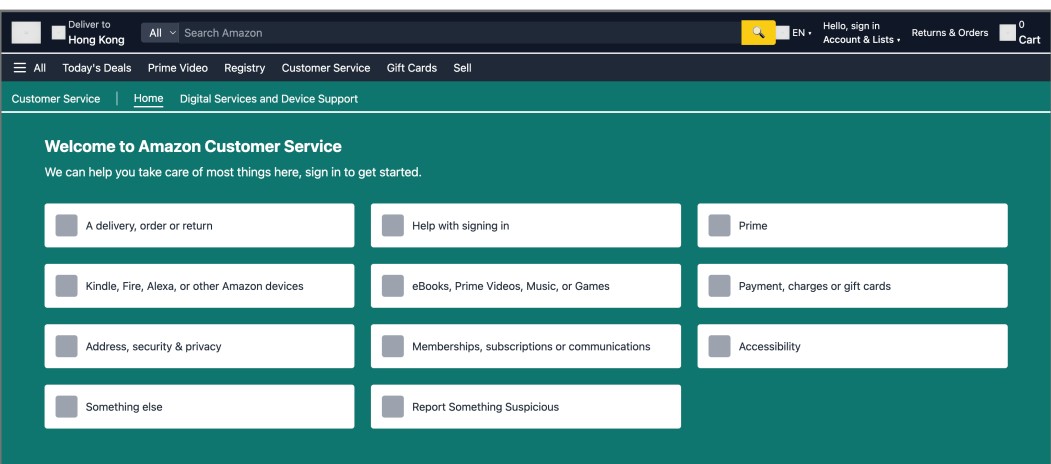

Generated by ScreenCoder (ours)

Figure 6: Qualitative comparison between our proposed method and various baselines.

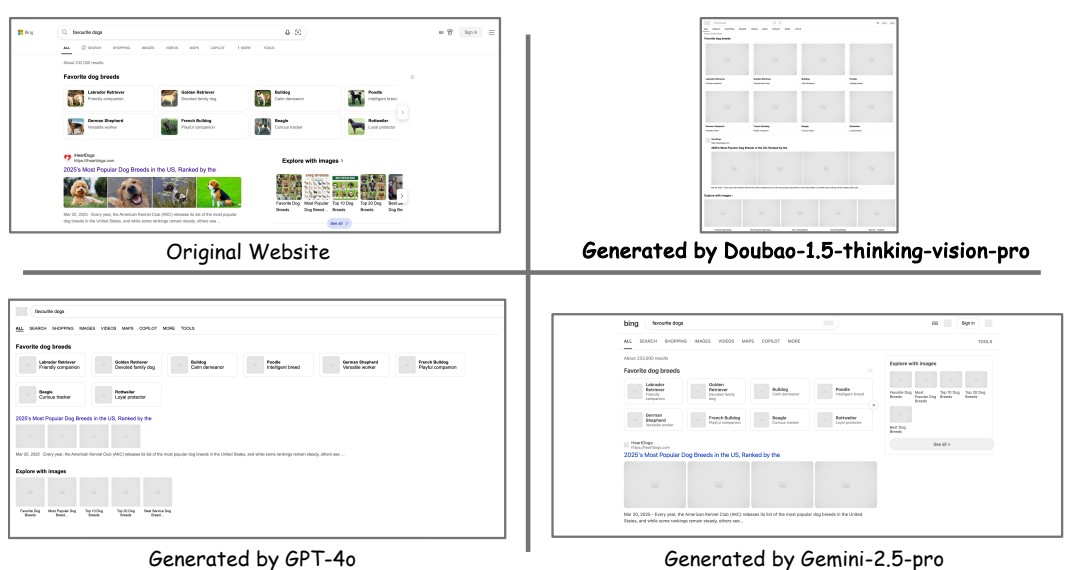

Figure 7: Qualitative comparison between our proposed method and various baselines.

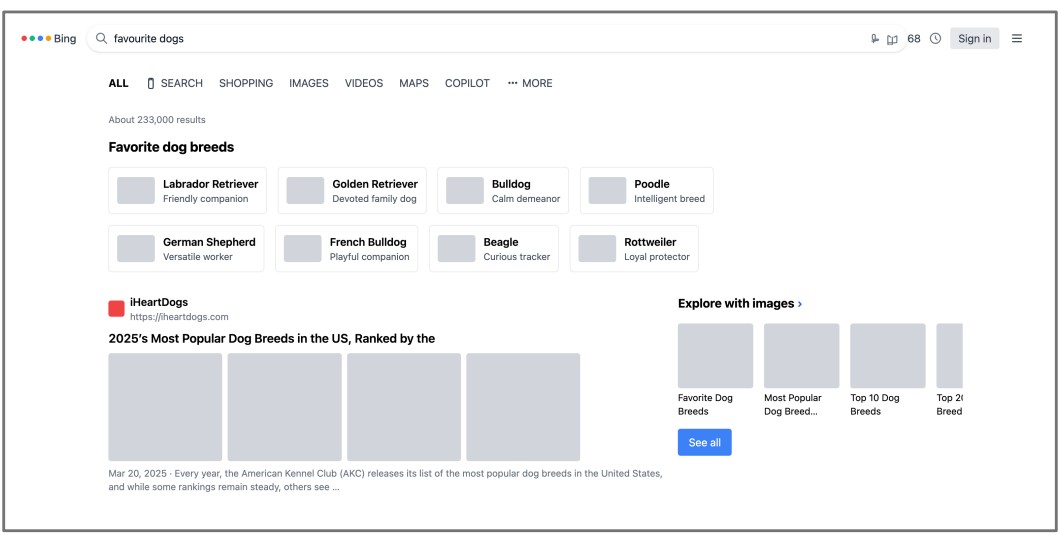

Generated by ScreenCoder (ours)

Figure 8: Qualitative comparison between our proposed method and various baselines.

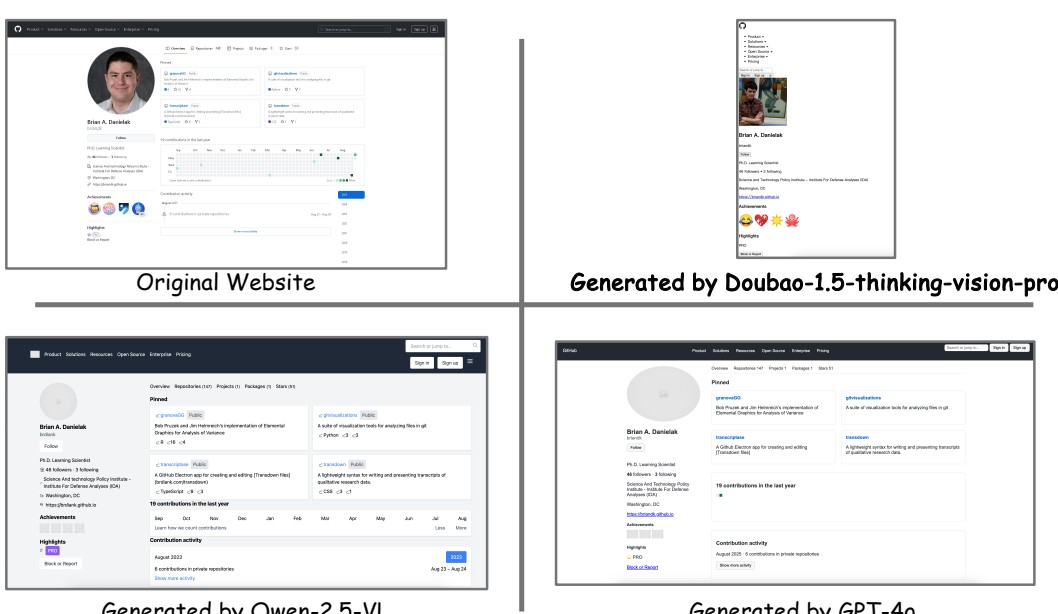

Original Website        Generated by Doubao-1.5-thinking-vision-pro

Generated by Qwen-2.5-VL        Generated by GPT-4o

Figure 9: Qualitative comparison between our proposed method and various baselines.

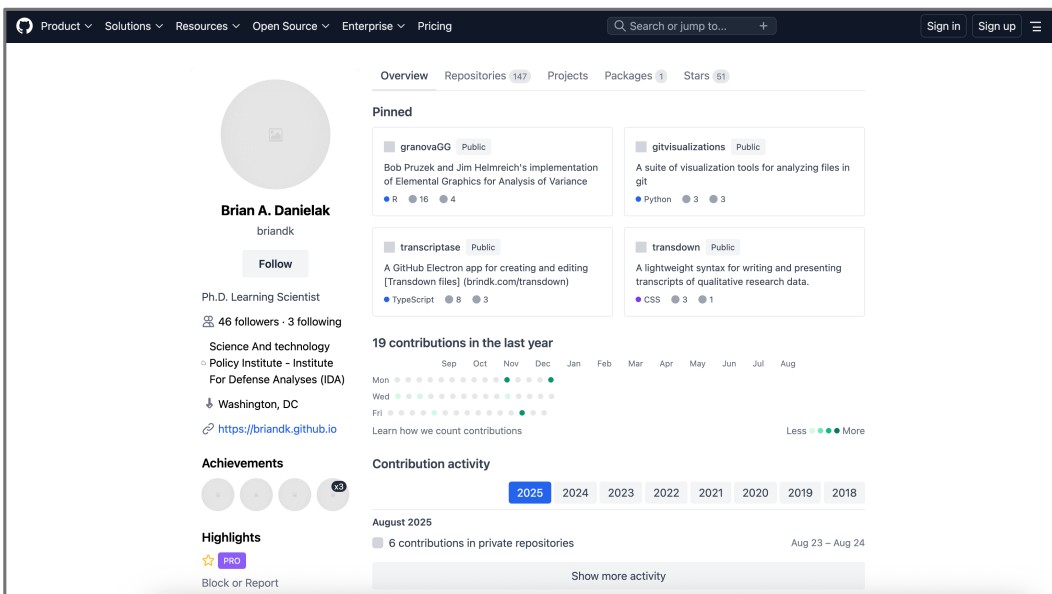

Generated by ScreenCoder (ours)

Figure 10: Qualitative comparison between our proposed method and various baselines.

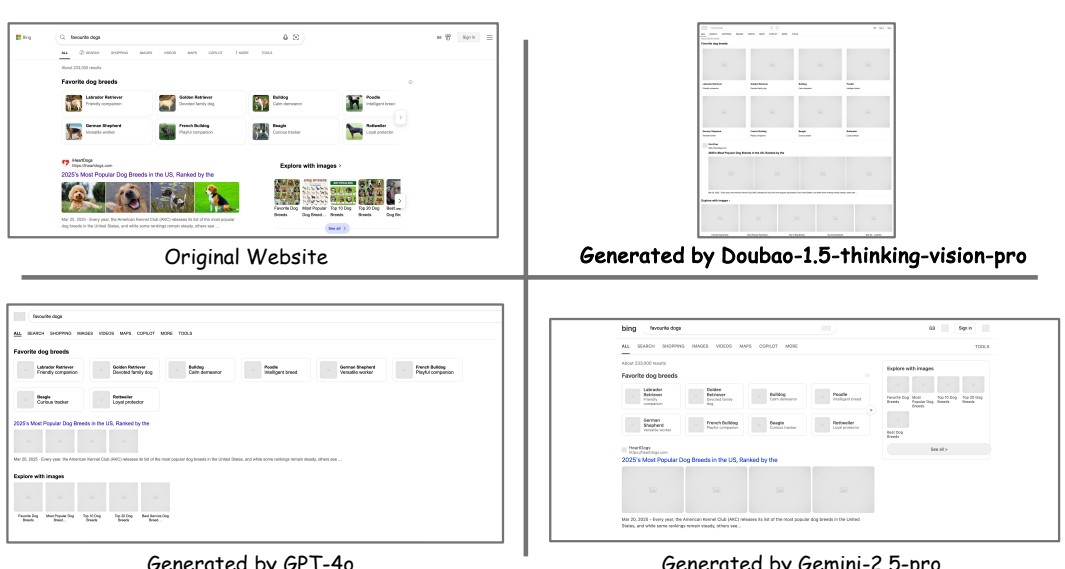

Original Website · Generated by Doubao-1.5-thinking-vision-pro

Generated by GPT-4o · Generated by Gemini-2.5-pro

Figure 11: Qualitative comparison between our proposed method and various baselines.

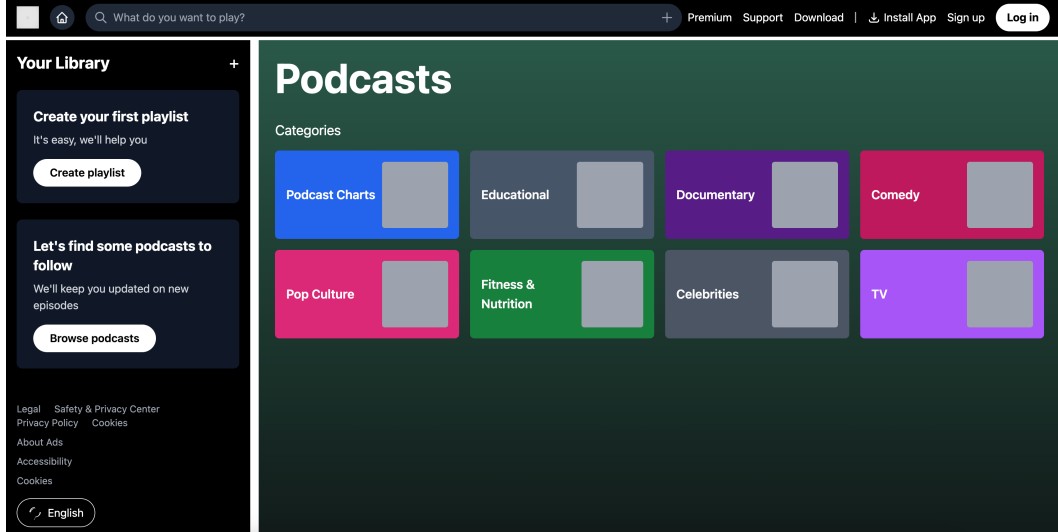

Generated by ScreenCoder (ours)

Figure 12: Qualitative comparison between our proposed method and various baselines.

## D  LLM Usage Statement

We used LLMs as an assistive tool for improving grammar and clarity in the text, and for code completion. The core research, including the experimental design and final implementation, was conceived and executed entirely by the authors.

## E  Failure Case Analysis and Robustness Check

In this section, we provide a qualitative analysis of failure modes, specifically focusing on "Hard Cases" involving non-standard, artistic layouts that deviate from the standard rectangular box model. We compare the performance of ScreenCoder against the strong end-to-end baseline, Qwen2.5-VL.

As illustrated in Figures 15 to 17, we test the models on a design featuring significant CSS transformations (e.g., `skewY`) and overlapping elements. This analysis highlights a critical trade-off between pixel imitation and structural integrity:

- **Catastrophic Failure in Baselines (Qwen2.5-VL):** Monolithic MLLMs tend to prioritize pixel-level visual matching. When encountering skewed or rotated elements, the baseline attempts to replicate the geometry using absolute positioning and incorrect transform approximations. This results in "spatial hallucination," where elements suffer from Z-index errors, text overlaps, and fragmented coordinate placement. The resulting code, while attempting to look like the input, is functionally unusable.

- **Graceful Degradation in ScreenCoder (Ours):** Our modular approach prioritizes valid DOM structure. The Planning Agent maps the non-rectangular visual input to the nearest robust rectangular grid (CSS Flexbox/Grid). While this results in a loss of the specific artistic nuance (the skew angle is removed), the system fails *gracefully*. The generated code remains clean, responsive, and structurally valid. This demonstrates that ScreenCoder prefers simplification over broken complexity, ensuring the output is always a viable starting point for developers.

**Prompt for Grounding**

**Seed-1.5-thinking-pro-vision**    """Only return the bounding boxes of the 'header', 'footer', 'main content', 'sidebar', and 'navigation' in this webpage screenshot. Please only return the corresponding bounding boxes in the format of 'name: <bbox>x1 y1 x2 y2</bbox>' for each line without extra information and comments!!! Note: 1. All text information, images and other content should be framed, don't miss any information; 2. The areas should not overlap; 3. Output a label and the corresponding bounding box for each line. You can decide whether to include some regions or not."""

**Qwen-2.5-VL**    """Only return the bounding boxes of the 'header', 'footer', 'main content', 'sidebar', and 'navigation' in this webpage screenshot. Please only return the corresponding bounding boxes in the format of 'header: <bbox>x1 y1 x2 y2</bbox>\nfooter: <bbox>x1 y1 x2 y2</bbox>\nmain content: <bbox>x1 y1 x2 y2</bbox>\nsidebar: <bbox>x1 y1 x2 y2</bbox>\nnavigationr: <bbox>x1 y1 x2 y2</bbox>\n 'for each line without extra information and comments!!! Note: 1. All text information, images and other content should be framed, don't miss any information; 2. The areas should not overlap; 3. Output a label and the corresponding bounding box for each line. You can decide whether to include some regions or not. Also, output the image size in the format of "width: <width> height: <height>\n"."""

Figure 13: Prompt Templates.

1080
1081
1082
1083
1084
1085
1086
1087
1088
1089
1090
1091
1092
1093
1094
1095
1096
1097
1098
1099
1100
1101
1102
1103
1104
1105
1106
1107
1108
1109
1110
1111
1112
1113
1114
1115
1116
1117
1118
1119
1120
1121
1122
1123
1124
1125
1126
1127
1128
1129
1130
1131
1132
1133

**Prompt for Generation**

{"sidebar": f"""This is a screenshot of a container. Here is the user's additional instruction: {user_instruction["sidebar"]}. Please fill in a complete HTML and Tailwind CSS code to accurately reproduce the given container. Please ensure that all block layouts, icon styles, sizes, and text information are consistent with the original screenshot, based on the user's additional conditions. Below is the code template to fill in:
<div>
your code here
</div>
Only return the code within the <div> and </div> tags.""",

"header": f"""This is a screenshot of a container. Here is the user's additional instruction: {user_instruction["header"]}. Please fill in a complete HTML and Tailwind CSS code to accurately reproduce the given container. Please ensure that all blocks' relative positions, layout, text information, and colors within the bounding box are consistent with the original screenshot, based on the user's additional conditions. Below is the code template to fill in:
<div>
your code here
</div>
Only return the code within the <div> and </div> tags.""",

"navigation": f"""This is a screenshot of a container. Here is the user's additional instruction: {user_instruction["navigation"]} Please fill in a complete HTML and Tailwind CSS code to accurately reproduce the given container. Please ensure that all blocks' relative positions, text layout, and colors within the bounding box are consistent with the original screenshot, based on the user's additional conditions. Please use the same icons as in the original screenshot. Below is the code template to fill in:
<div>
your code here
</div>
Only return the code within the <div> and </div> tags.""",

"main content": f"""This is a screenshot of a container. Here is the user's additional instruction: {user_instruction["main content"]} Please fill in a complete HTML and Tailwind CSS code to accurately reproduce the given container. Please replace the images in the original screenshot with solid gray blocks of the same size; text inside the images does not need to be recognized. Please ensure that all blocks' relative positions, layout, text information, and colors within the bounding box are consistent with the original screenshot, based on the user's additional conditions. Below is the code template to fill in:
<div>
your code here
</div>
Only return the code within the <div> and </div> tags.""",

"footer ": """This is a screenshot of a container. Here is the user's additional instruction: {user_instruction["footer"]}. Please fill in a complete HTML and Tailwind CSS code to accurately reproduce the given container. Please ensure that all blocks' relative positions, text layout, and colors within the bounding box are consistent with the original screenshot, based on the user's additional conditions. Please use the same icons as in the original screenshot. Below is the code template to fill in:
<div>
your code here
</div>
Only return the code within the <div> and </div> tags."""}

user_instruction: dictionary for storing user-defined instructions for each region ("header", "footer", "sidebar", "navigation", "main content").

Figure 14: Prompt Templates.

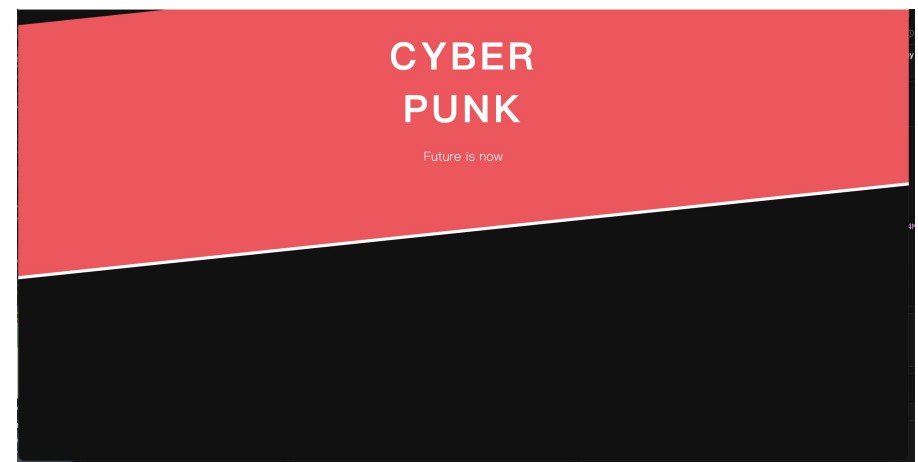

Figure 15: **Failure Case Part 1: Input Design.** The input features a non-standard layout with a skewed container and rotated text.

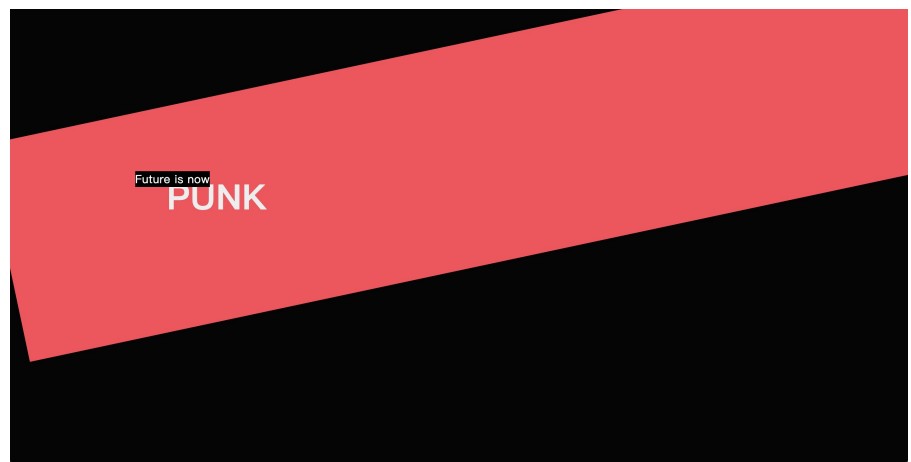

Figure 16: **Failure Case Part 2: Qwen2.5-VL (Baseline).** The baseline suffers from *spatial hallucination*. It prioritizes pixel matching via fragile absolute positioning, leading to Z-index errors and overlapping text.

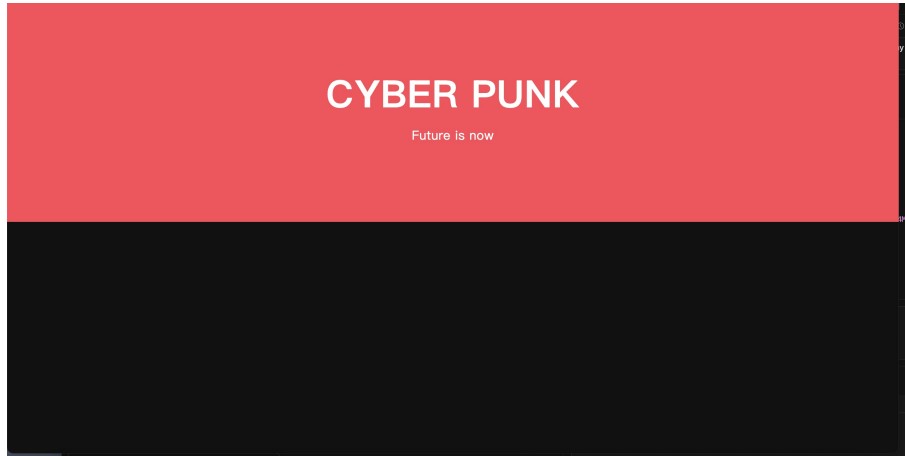

Figure 17: **Failure Case Part 3: ScreenCoder (Ours).** ScreenCoder simplifies the geometry to a standard rectangular grid (Graceful Degradation). While it misses the artistic skew, it maintains structural integrity, generating clean and valid code.

