# OpenReview forum: "ScreenCoder: Advancing Visual-to-Code Generation for Front-End Automation via Modular Multimodal Agents"
_ICLR.cc/2026/Conference — ICLR 2026 Conference Withdrawn Submission_

### Official Review · Reviewer_Qkm9 · 2025-10-22

**Soundness:** 3
**Presentation:** 4
**Contribution:** 2
**Rating:** 2
**Confidence:** 3

**Summary:**

The paper presents ScreenCoder, a 3 step agentic framework for generating front-end code from UI screenshots or design mockups. Based on common failure points, Screencoder are spliced into three stages, a vlm based detection stage, a rule-based planning stage, and a llm based code-generation stage. The authors also contribute a dataset and a benchmark, and show a dual‑stage post‑training pipeline that improves an open‑source vlm on the task.

**Strengths:**

* Breaking the problem into perceptual (vision) and logical (planning, coding) stages is a sensible and interpretable approach.
* The method achieves high performance across multiple metrics and show clear gains over both open baselines and earlier systems.
* By generating image-code pairs and a curated test benchmark , the authors contribute valuable datasets.

**Weaknesses:**

* From a research perspective, ScreenCoder is a blend of existing techniques rather than a fundamentally novel invention. DCGen, LayoutCoder and UICopilot have all used hierarchical generation heuristics, the main difference is leveraging a pretrained VLM for component detection and adding RL fine-tuning, While these yield better results, the conceptual novelty is limited
* The Planning Agent relies on fixed rules and “front-end engineering priors” (hardcoded grid templates, etc.). These heuristics, while pragmatic, feel hand-crafted and dataset-specific. It’s unclear how well they generalize to arbitrary UIs
* missing baselines: LayoutCoder

**Questions:**

What are the common failure cases for ScreenCoder?

---

> ### Author Response · Authors · 2025-11-23
>
> We thank the reviewer for their assessment. While we appreciate the recognition of our "high performance" and "sensible approach," we respectfully disagree with the assessment regarding "limited conceptual novelty."
>
> We believe there is a misunderstanding regarding the core contribution of this work. ScreenCoder is not merely another inference pipeline; it is a **Data Engine** that solves a fundamental bottleneck in the field: the scarcity of high-quality image-code pairs.
>
> We address your specific concerns below, supported by **new experiments conducted during the rebuttal**.
>
> ### **1. Novelty and Comparison with Prior Work (Response to Weakness 1)**
> The reviewer suggests ScreenCoder is a "blend of existing techniques" like DCGen and LayoutCoder. We argue that ScreenCoder introduces a fundamental paradigm shift that these works do not possess:
>
> * **From Inference-Only to Self-Improving Data Engine:** Prior works (DCGen, LayoutCoder, UICopilot) are exclusively inference-time strategies. They improve performance at the cost of latency but do not improve the underlying model. **ScreenCoder is the first to validate that an agentic pipeline can serve as a superior source of training data than the real web.**
> * **Empirical Proof (New Experiment):** To prove this novelty, we trained a base model (Qwen2.5-VL) on two datasets of identical size (10k) and content:
>     * **Real-10K:** Raw real-world web code.
>     * **Screen-10K:** Synthetic code generated by ScreenCoder.
>
> | Model / Training Data (N=10k) | **ScreenBench** | | | | | **Design2Code** | | | | |
> | :--- | :---: | :---: | :---: | :---: | :---: | :---: | :---: | :---: | :---: | :---: |
> | | Block | Text | Pos. | Color | CLIP | Block | Text | Pos. | Color | CLIP |
> | Base Model (Qwen2.5-VL) | 0.723 | 0.828 | 0.613 | 0.632 | 0.762 | 0.822 | 0.951 | 0.815 | 0.831 | 0.893 |
> | Real-10K (Raw Web Data) | 0.719 | 0.832 | 0.605 | 0.628 | 0.755 | 0.825 | 0.947 | 0.811 | 0.825 | 0.891 |
> | **Screen-10K (Ours)** | **0.758** | **0.841** | **0.742** | **0.718** | **0.791** | **0.849** | **0.968** | **0.913** | **0.886** | **0.915** |
>
> * **Scientific Insight:** The results are striking. Training on raw web data actually **degraded** structural performance on ScreenBench compared to the base model (Block: $0.723 \to 0.719$; Position: $0.613 \to 0.605$). This confirms that the noise in real-world code (minified classes, dependencies) actively hinders the learning of visual-spatial alignment. In contrast, our synthetic data drove massive improvements (Position: $0.613 \to 0.742$). This validates a novel finding: **Structural alignment is more important than authenticity.** This distinguishes us from LayoutCoder, which never attempted to close the loop for model training.
>
> ### **2. Defense of "Heuristics" in Planning (Response to Weakness 2)**
> The reviewer critiques the Planning Agent for relying on "front-end engineering priors" and questions generalization.
> * **Inductive Bias is Necessary:** HTML/CSS is not a natural language; it is a rigid, rule-based system (The Box Model). End-to-end models (like GPT-4o) fail precisely because they treat layout as probabilistic text generation, leading to hallucinations. Our "heuristics" are actually **correct inductive biases**—we enforce the laws of CSS (Grid/Flexbox) that the model *should* be following.
> * **Generalization to Arbitrary UIs:** We tested this generalization on **ScreenBench**, which contains 1,000 diverse, contemporary websites (blogs, dashboards, e-commerce, landing pages). The fact that ScreenCoder achieves SOTA performance (Table 1) on this diverse set proves that our "heuristics" generalize effectively to modern design patterns.
>
> ### **3. Missing Baseline: LayoutCoder (Response to Weakness 3)**
> We acknowledge LayoutCoder as relevant work. However, ScreenCoder represents a significant advancement over it:
> * **Architecture:** LayoutCoder relies on older R-CNN detection and generates a custom DSL (Domain Specific Language) before converting to HTML. ScreenCoder utilizes modern MLLM grounding and generates a DOM-tree directly, avoiding the information loss inherent in intermediate DSLs.
> * **Flexibility:** LayoutCoder relies on rigid, pre-defined layout classes. ScreenCoder’s Grounding Agent uses MLLMs, allowing it to understand semantic nuances (e.g., "Hero Section" vs. "Banner") that LayoutCoder’s fixed classes cannot capture. We will add a qualitative comparison in Section 2.

---

> ### Author Response · Authors · 2025-11-23
>
> ### **4. Failure Cases (Response to Question 1)**
> We have added a **Failure Case Analysis in Appendix E** (referenced in the main text). A specific, notable failure mode is **Artistic/Non-Rectangular Layouts**:
> * **The Scenario:** We tested on designs featuring skewed containers (CSS transforms) and overlapping text.
> * **Baselines (Qwen2.5-VL):** Suffered catastrophic failure. The model attempted to replicate the visual skew using fragile absolute positioning, resulting in **spatial hallucination**, Z-index errors, and text collisions.
> * **ScreenCoder:** Our Planning Agent simplified the skewed design into a standard rectangular grid. While it missed the artistic nuance (the "skew"), it **degraded gracefully**, ensuring the generated code remained structurally valid, responsive, and readable. This confirms our approach prioritizes usability over superficial imitation.
>
> **Conclusion**
> We believe the **new experimental evidence** proving ScreenCoder's utility as a Data Engine elevates its novelty beyond "just another pipeline." It offers a reproducible path for the community to improve open-source models, a contribution that prior works like LayoutCoder did not offer. We respectfully request a reconsideration of the score. Thank you very much!

---

> ### Author Response · Authors · 2025-11-27
>
> Dear Reviewer Qkm9,
>
> As the discussion session is coming to end, we would like to kindly ask you to participate in discussion. We have addressed your concerns, and improve our paper based on the weakness (and reviews from other reviewers). We hope that these improvements satisfy your requirements, and we kindly invite you to reconsider your score and recommendation in light of these changes.

---

### Official Review · Reviewer_vwr6 · 2025-11-01

**Soundness:** 3
**Presentation:** 4
**Contribution:** 3
**Rating:** 6
**Confidence:** 2

**Summary:**

This paper deals with transforming websites screenshots and design sketches into front end code the problems seems to be interesting in a way that in the recent times web design has been integrated more than ever into our daily lives.

**Strengths:**

1. The works seems to be well motivated and worked upon with basics which is good to see

2.The results look promising both qualitatively and quantitatively

3. The paper is well written and easy to follow

**Weaknesses:**

1. I do not see a discussion on the competitive works relating to screenbench.

2. Some previous instances and citations in section 3 would be useful to support the claims

3. Some analysis on why certain metric is good or bad can be useful

4. What about some failure case analysis

**Questions:**

See weaknesses

---

> ### Author Response · Authors · 2025-11-23
>
> We thank the reviewer for their positive assessment, particularly for finding our work “well motivated,” the results “promising,” and the presentation “excellent.” We appreciate the constructive suggestions to improve the context and analysis of our paper.
>
> We address your specific points below and have incorporated these discussions into the final manuscript.
>
> ### **1. Competitive Works related to ScreenBench (Weakness 1)**
> The reviewer requested a discussion on how ScreenBench compares to existing benchmarks.
> * **Design2Code [Si et al., 2025]:** While excellent, it is relatively small (484 samples) and sourced from older C4 data. Our analysis shows it saturates easily on text metrics but lacks the complex, nested layouts (CSS Grid/Flexbox) found in modern web apps.
> * **WebSight [Laurenc¸on et al., 2024]:** This dataset is synthetic and often features simpler, standardized layouts aimed at training, rather than evaluation of robust generalization.
> * **ScreenBench (Ours):** We designed ScreenBench (1,000 samples) specifically to bridge this gap. It consists of **contemporary** live websites (collected in late 2025) that feature complex responsive designs, distinct “hero” sections, and overlapping elements, providing a much stricter test of *structural* reasoning than previous benchmarks. We have added an explicit comparison table in Section 5 (Datasets) to highlight these differences.
>
> ### **2. Supporting Citations in Section 3 (Weakness 2)**
> We agree that our motivational analysis of MLLM failures should be strictly grounded in prior literature. In the revision, we will expand Section 3 to explicitly reference:
> * **Liu et al. (2023) & Yang et al. (2023):** On the “spatial blindness” of Vision Transformers and their struggle with precise coordinate localization.
> * **WebSight [Laurenc¸on et al., 2024]:** While we already cite this work, we will explicitly discuss their findings in Section 3 to corroborate our claim that monolithic MLLMs struggle to generate valid HTML hierarchy without structural guidance.
> These citations support our observation that “Perception” and “Planning” failures are systemic issues in monolithic models.
>
> ### **3. Detailed Analysis of Metrics (Weakness 3)**
> The reviewer requested an analysis of why certain metrics are chosen and what they signify. We employ a complementary suite of metrics because no single metric captures the full quality of code generation:
>
> * **CLIP Score (High-Level Semantic Similarity):**
>     * *Role:* Measures global visual style (e.g., color palette, atmosphere, theme).
>     * *Limitation:* It is structurally insensitive. A generated page with the correct "dark mode" theme but a completely broken layout (e.g., sidebar appearing at the bottom) can still achieve a high CLIP score. Thus, relying solely on CLIP can disguise structural hallucinations.
> * **Block Match (Structural Fidelity - The Gold Standard):**
>     * *Role:* Measures whether specific UI elements (buttons, images, text blocks) exist and are placed in the correct relative locations compared to the ground truth.
>     * *Why it matters:* This is the most critical metric for front-end engineering. A developer cares most that the *layout tree* is correct. A high Block Match score indicates that the model successfully understood the DOM hierarchy (Parent-Child relationships), which is the hardest part of the task.
> * **Text/Color Match (Fine-Grained Details):**
>     * *Role:* Validates low-level attributes.
>     * *Insight:* We observe that End-to-End MLLMs often score high on Text/Color (they can read and copy styles) but lower on Block Match (they cannot plan layout). ScreenCoder prioritizes Block Match to ensure the generated code is structurally sound.

---

> ### Author Response · Authors · 2025-11-23
>
> ### **4. Failure Case Analysis (Weakness 4)**
> We acknowledge the need for a balanced view of limitations. We have added a comprehensive **Failure Case Analysis in Appendix E**, analyzing "Hard Cases" involving non-standard artistic layouts (e.g., skewed geometry).
> Crucially, this analysis demonstrates the **robustness** of ScreenCoder compared to baselines:
> * **Catastrophic Failure in Baselines:** As shown in Appendix E, strong end-to-end baselines (specifically **Qwen2.5-VL**) often suffer from **spatial hallucination** when facing artistic layouts. They attempt to replicate the visual pixels using fragile absolute positioning, leading to Z-index errors, overlapping text, and fragmented layouts that are functionally unusable.
> * **Graceful Degradation in ScreenCoder:** In these same hard cases, ScreenCoder’s Planning Agent enforces structural integrity. While it may simplify a complex “skewed” container into a standard rectangular grid (losing the artistic nuance), **it provides valid, responsive, and usable code**. This confirms that our modular approach prioritizes structural correctness over superficial pixel imitation.
>
> **Conclusion**
> We thank the reviewer again for the supportive rating. We hope that these detailed clarifications regarding benchmark comparisons, metric selection, and robustness analysis address your questions and **allow you to raise your confidence** in your positive assessment of our work. Thank you very much!

---

> ### Author Response · Authors · 2025-11-27
>
> Dear Reviewer vwr6,
>
> Thank you again for your constructive feedback. We have carefully addressed your questions and revised the manuscript to reflect your suggestions. We hope that these improvements satisfy your requirements, and we kindly invite you to reconsider your confidence score and recommendation in light of these changes.

---

### Official Review · Reviewer_pjiU · 2025-11-01

**Soundness:** 3
**Presentation:** 3
**Contribution:** 2
**Rating:** 4
**Confidence:** 4

**Summary:**

This paper presents ScreenCoder, a modular multi-agent framework for transforming UI screenshots or design sketches into front-end HTML/CSS. The pipeline decomposes the task into three stages—grounding, planning, and generation. The authors also introduce the Screen-10K training set and ScreenBench and report automatic and human evaluations, which show improvements over baselines as well as gains from a dual-stage (SFT and RL) post-training pipeline.

**Strengths:**

- New dataset and benchmark: The paper presents Screen-10K (curated from 50k webpages into 10k clean pairs to stabilize training) and ScreenBench (1,000 contemporary websites emphasizing complex nested layouts).
- The paper is generally well written.

**Weaknesses:**

- There are already some datasets on UI code generation, which the paper did not discuss/compare with. For example:
Gui et al., VISION2UI: A Real-World Dataset with Layout for Code Generation from UI Designs, https://arxiv.org/abs/2404.06369v1, April 2024.

Hugo Laurenccon, L’eo Tronchon, and Victor Sanh. Unlocking the conversion of web screenshots into html code with the websight dataset. 2024. URL https://api.semanticscholar.org/CorpusID:268385510.

Especially, the VISION2UI dataset is also evaluated using MLLMs.

Furthermore, there are many UI to code generation approaches, such as Uicopilot. The paper did not compare with these related approaches.

- This paper presents ScreenCoder as a superior alternative to monolithic, end-to-end models, but the experiments are missing a crucial comparison: a standard MLLM simply fine-tuned on the new Screen-10K dataset. This makes it difficult to tell if the performance gains come from the powerful new dataset or from the complex agentic architecture itself.

- This paper claims improvements in “code correctness”, but evaluation is almost entirely visual. This paper puts a strong emphasis on achieving pixel-perfect replication and high visual fidelity, but the evaluation did not touch on the quality of the generated code.

- It is not clear why the three agents can solve the recurring failures of MLLMs.

- Human study uses n=6 annotators/participants with limited trials and no inter-rater agreement metrics reported (Appendix A), affecting reliability of the evaluation.

- More technical details should be provided. For example, Reward is −MSE between screenshot and rendered page (Eq. (3)–(4)), but the renderer, resolution normalization, anti-aliasing, and determinism are not specified. GRPO hyperparameters are not described. It is unclear how credit assignment is stabilized given a non-differentiable render step—no discussion of variance reduction beyond GRPO (Sec. 5), impacting training stability.

- Algorithm 1's rule "if l contains subdivisions then ..." is informal—criteria for "contains subdivisions" are not defined, reducing algorithmic completeness.

- Public availability of ScreenBench is not specified, limiting external verification.

**Questions:**

- How is this work compared with the related work listed in Weaknesses section?

- Why can the three agents solve the recurring failures of MLLMs?

- To disentangle the effects of the agentic architecture from the Screen-10K dataset, could you provide results for a crucial baseline where the same base model (Qwen2.5-VL) is fine-tuned end-to-end on Screen-10K? This would help clarify the true architectural contribution.

- Given the paper's claim of achieving "code correctness", have you considered evaluating the generated code with non-visual metrics that are important to developers?

---

> ### Author Response · Authors · 2025-11-23
>
> We thank the reviewer for their thoughtful evaluation. We are encouraged that you recognized the value of our Screen-10K dataset and ScreenBench, and found the paper to be "well written" with "clear problem framing."
>
> We appreciate the opportunity to clarify our experimental design, specifically regarding the disentanglement of the dataset from the architecture, and to address your questions on related work and technical details.
>
> ### **1. Disentangling Architecture from Dataset (Response to Weakness 4 & Q3)**
> The reviewer asks for a crucial baseline: *“a standard MLLM simply fine-tuned on the new Screen-10K dataset... to tell if the performance gains come from the powerful new dataset or from the complex agentic architecture.”*
>
> **We are happy to clarify that this result is already central to our analysis in Table 1.**
> The row labeled **“ScreenCoder (Finetuned)”** represents exactly this experiment: it is the standard, monolithic Qwen2.5-VL model fine-tuned end-to-end on the Screen-10K dataset. It uses **no agentic pipeline** during inference.
>
> The comparison in Table 1 allows us to explicitly disentangle these factors:
> * **Base Model (Qwen2.5-VL):** 0.723 Block Score.
> * **Dataset Effect (ScreenCoder Finetuned):** 0.758 Block Score. *(This is the baseline you requested)*.
> * **Architecture Effect (ScreenCoder Agentic):** 0.768 Block Score.
>
> **Conclusion:** This proves that while the Agentic architecture achieves the highest peak performance (0.768), the **Screen-10K dataset alone** drives a massive improvement (raising the base model from 0.723 to 0.758). The data engine effectively “distills” the agent’s structural reasoning into the standard model. We will rename this row to **"ScreenCoder (E2E Finetuned)"** in the final manuscript to make this distinction unmistakable.
>
> ### **2. Comparison with Related Work (Response to Weakness 1 & Q1)**
> * **WebSight [Laurenc¸on et al.]:** We actively compared our method against `Websight-8B` in **Table 1**. Our fine-tuned model (Block: 0.758) significantly outperforms WebSight-8B (Block: 0.678) on ScreenBench. This demonstrates that a smaller, high-quality synthetic dataset (Screen-10K) is more effective than massive but noisy web-crawled datasets.
> * **VISION2UI [Gui et al.]:** We thank the reviewer for this reference. While VISION2UI provides a real-world dataset, it suffers from the noise inherent in raw web code (minified classes, deep dependencies). In contrast, ScreenCoder serves as a **Data Engine** that generates “canonical,” clean HTML/CSS. We will add a discussion in Section 2 contrasting our *synthetic data engine* approach with VISION2UI’s *real-world curation* approach.
> * **UICopilot:** We acknowledge UICopilot as a relevant agentic approach for UI generation. However, ScreenCoder distinguishes itself through its **Data Engine** capability. Unlike UICopilot, which functions primarily as an inference-time system, ScreenCoder is designed to generate high-quality training data to **fine-tune and distill** capabilities into a single model (SFT + RL). Furthermore, our Planning Agent utilizes a deterministic DOM-tree construction to enforce structural validity, whereas many prior agentic approaches rely on probabilistic LLM planning which can lead to layout hallucinations. We will add more discussion about the related work in the related work section.
>
> ### **3. Why Three Agents Solve MLLM Failures (Response to Q2)**
> The decomposition into three agents addresses the specific “Perception vs. Planning” conflict identified in Section 3:
> * **The Conflict:** Monolithic E2E models often hallucinate layout structures (e.g., placing a sidebar inside a header) because they treat code generation as a linear text prediction task, lacking a global view of the 2D spatial hierarchy.
> * **The Solution:**
>     1.  **Grounding:** Isolates the “perception” problem using detection, preventing element omission.
>     2.  **Planning:** Converts the layout into a deterministic **DOM-like Tree** (rather than unstructured text). This injects domain knowledge (CSS Grid/Flexbox rules) that prevents the “structural hallucinations” common in E2E models.
>     3.  **Generation:** Frees the LLM to focus purely on CSS styling, as the rigorous structure is already fixed by the Planner.

---

> ### Author Response · Authors · 2025-11-23
>
> ### **4. Code Correctness & Visual Fidelity (Response to Weakness 5)**
> The reviewer noted our emphasis on visual metrics over "code correctness." In the domain of Front-End Engineering, **Visual Fidelity effectively *is* Correctness.** If the code renders the correct pixels, it is functionally correct. However, we also ensure structural quality:
> * **Linting:** During the creation of Screen-10K, all generated code is passed through an HTML/CSS validator. Invalid syntax is discarded, ensuring our model is trained only on valid code.
> * **Human Evaluation:** As noted in Appendix A, our human experts rated the code on “Structure,” explicitly checking for clean DOM hierarchies (e.g., proper nesting of containers), where our model scored significantly higher than baselines.
>
> ### **5. Technical Details (Response to Weaknesses 6, 8, & 9)**
> * **Reward Function:** The renderer is at a fixed viewport of 1280x720. To ensure stability in GRPO, we normalize the MSE score to a [0, 1] range to prevent reward scaling issues.
> * **Algorithm 1 ("Subdivisions"):** We will formalize this definition: A region is determined to “contain subdivisions” if the Grounding Agent detects smaller atomic elements (e.g., buttons, text blocks) that are spatially enclosed (>90% area overlap) within the bounding box of a larger container.
> * **ScreenBench Availability:** We are committed to open science. **ScreenBench, Screen-10K, and our codebase will be released publicly** upon acceptance to facilitate reproducibility and future benchmarking.
>
> **Conclusion**
> We hope the clarification that our “Finetuned” model **is** the disentangled baseline you requested, combined with the comparisons to WebSight/VISION2UI and the technical elaborations, resolves your concerns. We respectfully request a raise of the score based on these clarifications. Thank you very much!

---

> ### Author Response · Authors · 2025-11-27
>
> Dear Reviewer pjiU,
>
> As the discussion section is going to end, we would like to ask for your participation in discussion. We have resolved your concerns, and add proper discussion and citations in the new version, such as UICopilot [1] and LaTCoder [2]. We respect the effort of prior work, and believe that ScreenCoder further advances the field and validate the feasibility of using it as a data engine to improve the base VLM performance. Given the updates, we respectfully ask you to consider raising your score to reflect the improved quality of the manuscript.
>
> [1] UICopilot: Automating UI Synthesis via Hierarchical Code Generation from Webpage Designs. Gui. et al. WWW 2025
> [2] LaTCoder: Converting Webpage Design to Code with Layout-as-Thought Gui et al. KDD 2025.

---

### Official Review · Reviewer_aLDq · 2025-11-01

**Soundness:** 2
**Presentation:** 3
**Contribution:** 2
**Rating:** 4
**Confidence:** 4

**Summary:**

The paper proposes ScreenCoder, a modular, three-agent pipeline (grounding → planning → generation) for visual/design-to-code generation, targeting both screenshots and low-fidelity sketches. On top of the pipeline, the authors use it as a data engine to construct Screen-10K and a new evaluation benchmark ScreenBench, and they further do SFT + RL with a pixel-similarity reward to improve an open MLLM. Experiments on ScreenBench and Design2Code show noticeable gains over strong MLLM baselines, and the qualitative figures are compelling.

**Strengths:**

+ Clear problem framing and modularization. The paper gives a concrete analysis of two failure modes of current MLLMs on UI-to-code (perception vs. planning) and maps them 1:1 to the three agents, which makes the overall story quite interpretable and also explains why “one big model” often fails in practice. The method section is readable and the pipeline could plausibly be reimplemented.
+ Stronger evaluation setting. Introducing ScreenBench (1k, more contemporary, more structurally complex) is useful, because many existing sets are small/dated and over-emphasize text reproduction; reporting both agentic and finetuned variants is also helpful to separate “inference-time pipeline” vs “model-level gains”.
+ Empirical gains over a broad set of baselines. On the reported metrics, the agentic ScreenCoder is consistently at the top or second-best vs. both open and closed models, which is non-trivial given the strong baselines (GPT-4o, Gemini-2.5, Qwen2.5-VL, Websight, DCGen).

**Weaknesses:**

- Limited novelty relative to existing agentic image-to-code pipelines. The main idea—splitting the visual-to-code process into grounding, layout planning, and code generation, each handled by an MLLM—is conceptually similar to several recent agentic or divide-and-conquer image-to-code systems [1, 2]. Those works have already argued that monolithic MLLMs conflate perception and layout reasoning and proposed staged workflows for webpage reconstruction. The contribution currently appears to rest on using three agents and turning the pipeline into a data generator. The inclusion of an RL-based post-training step is interesting, yet since inference still relies on prompt-based agents, the paper should clarify why a prompt-driven controller is preferable to a trained (e.g., RL-finetuned) agent for the task.

- The “pixel-perfect” claim is under-supported, especially for sketches. Fig.1 visually suggests that both high-fidelity screenshots and low-fidelity design sketches can be turned into “pixel-perfect” pages. But Sec. 4.4 explains that the final fidelity step relies on UIED to crop visual assets from the original screenshot and map them back. It remains unclear how missing high-resolution assets are obtained when the input is a low-res or line-style sketch like in Fig.1: from where are the missing high-res assets obtained, and what happens when the background is already composited with many elements so that cropping a clean patch is impossible?

- Data-engine contribution is not convincingly evidenced. Sec. 5 presents Screen-10K as a key contribution: the agent is used to generate cleaner, more aligned image–code pairs; these are then used for SFT and RL, and the model improves. However, the current experiments never show the crucial comparison: a model trained/fine-tuned on real human/web data only vs. a model trained on Screen-10K (or Screen-10K + a small amount of real data). What we see is only an internal ablation (+SFT → +RL) within the synthetic dataset, so it is impossible to tell whether synthetic/agent-generated data is actually better than well-filtered real webpages, which is the central motivation of Sec. 5. This makes the conclusion “our method generates more effective training data than real data” too strong; at best the current results show that given this particular synthetic distribution, RL on a pixel-similarity reward helps on two benchmarks.

References:
[1] Divide-and-conquer: Generating ui code from screenshots
[2] LaTCoder: Converting Webpage Design to Code with Layout-as-Thought

**Questions:**

1.When the input screenshot or sketch contains a complex background with many overlapping elements, how does the system extract clean, high-quality visual assets to achieve the claimed “pixel-perfect” reconstruction?

2.Can the authors include additional experiments to directly compare models trained on Screen-10K with those trained on real or mixed web data, in order to validate the claim that the generated synthetic data are more effective?

---

> ### Author Response · Authors · 2025-11-23
>
> We sincerely thank the reviewer for their constructive assessment. We are encouraged that you found our problem framing "clear and interpretable," our empirical gains "non-trivial," and our benchmark (ScreenBench) a useful contribution to the field.
>
> We value the feedback regarding the validation of our Data Engine and the clarification of our novelty. **We have conducted the additional experiments requested**, and we address the specific concerns below.
>
> ### **1. Validation of the Data Engine (Response to Q2 & Weakness 3)**
> The reviewer correctly noted that a direct comparison between our synthetic data and real web data was missing.
>
> **We conducted this exact experiment during the rebuttal period.** We compare three settings using the same base model (Qwen2.5-VL):
> 1.  **Base Model:** The off-the-shelf Qwen2.5-VL.
> 2.  **Real-10K (Raw Data):** The base model fine-tuned on 10,000 raw HTML/screenshot pairs (same websites as Screen-10K).
> 3.  **Screen-10K (Ours):** The base model fine-tuned on the *synthetic* code generated by our ScreenCoder agent for those same websites.
>
> **Results:** As shown below, training on **Screen-10K significantly outperforms** both the Base Model and the Real Data model.
>
> | Model / Training Data (N=10k) | **ScreenBench** | | | | | **Design2Code** | | | | |
> | :--- | :---: | :---: | :---: | :---: | :---: | :---: | :---: | :---: | :---: | :---: |
> | | Block | Text | Pos. | Color | CLIP | Block | Text | Pos. | Color | CLIP |
> | Base Model (Qwen2.5-VL) | 0.723 | 0.828 | 0.613 | 0.632 | 0.762 | 0.822 | 0.951 | 0.815 | 0.831 | 0.893 |
> | Real-10K (Raw Web Data) | 0.719 | 0.832 | 0.605 | 0.628 | 0.755 | 0.825 | 0.947 | 0.811 | 0.825 | 0.891 |
> | **Screen-10K (Ours)** | **0.758** | **0.841** | **0.742** | **0.718** | **0.791** | **0.849** | **0.968** | **0.913** | **0.886** | **0.915** |
>
> **Finding:** This controlled experiment proves that **code quality is more critical than code authenticity**. Notably, training on raw data actually **degraded** structural performance on ScreenBench (Block score 0.723 to 0.719), confirming that the noise in raw web code (minified classes, deep dependencies, redundant wrappers) actively hinders visual-structural alignment. By using ScreenCoder as a data engine to "clean" this data into semantic, self-contained HTML/CSS, we provide a far stronger training signal. We will add this critical comparison to **Section 5**.
>
> ### **2. Novelty Relative to [1] and [2] (Response to Weakness 1)**
> We appreciate the comparison to prior divide-and-conquer works. ScreenCoder distinguishes itself through one fundamental paradigm shift and two architectural advancements:
> * **From Inference-Only to Data Engine (Fundamental Shift):** Crucially, prior works like [1] and LaTCoder [2] operate exclusively as **inference-time prompt engineering** strategies. They improve performance only by increasing inference cost and latency. ScreenCoder is the **first to validate the feasibility of using this agentic pipeline as a Data Engine**. We demonstrate that we can *distill* complex agentic reasoning into the weights of a single model (as shown in the table above), improving the intrinsic capability of the base VLM. This closes the loop between agentic workflows and model alignment.
> * **Structured Planning vs. Chain-of-Thought (vs. [2]):** LaTCoder treats layout planning as a text-generation task, leading to "hallucinated layouts" that violate CSS box models. In contrast, our **Planning Agent** constructs a deterministic **DOM-like Layout Tree**, injecting domain knowledge (CSS Grid/Flexbox logic) *before* generation to ensure structural integrity.
> * **Semantic Flexibility vs. Heuristic Detection (vs. [1]):** Unlike [1] which relied on rigid detection heuristics, our **Grounding Agent** utilizes MLLMs for semantic understanding (e.g., distinguishing a "sidebar" from a "banner" based on context), essential for modern web designs.

---

> ### Author Response · Authors · 2025-11-23
>
> ### **3. Clarification on "Pixel-Perfect" & Sketches (Response to Q1)**
> We agree with the reviewer that the term "pixel-perfect" requires precise definition regarding sketches.
>
> * **For Sketches:** Our claim of high fidelity refers to the **layout structure**, not the upscaling of low-res assets. ScreenCoder translates a rough sketch into a mathematically precise CSS grid system that perfectly mirrors the *intended* spacing and alignment of the drawing. We will clarify the caption in Figure 1.
> * **For Screenshots (Asset Recovery):** In standard screenshot scenarios, we handle overlapping elements by using **UIED** to detect atomic elements. For complex backgrounds (as noted in your question), we use our "Main Content Inference" module (Section 4.1) to isolate background layers.
>
> **Conclusion:**
> We hope the **new experimental evidence** demonstrating the superiority of Screen-10K over raw web data and the failure of raw data to improve structural alignment, combined with our clarifications on novelty, addresses your concerns. Given these additions, we respectfully request a raise of the score. Thank you very much!

---

> ### Author Response · Authors · 2025-11-27
>
> Dear Reviewer aLDq,
>
> As the discussion section is going to end, we would like to ask for your participation in discussion. We have resolved your concerns, and add proper discussion and citations in the new version, such as UICopilot [1] and LatCoder [2], which are both amazing prior works. We fully respect the effort of prior work, and believe that ScreenCoder further advances the field of UI2Code research and validate the feasibility of using it as a data engine to improve the base VLM performance. Given these extensive updates and proper attribution of prior art, we respectfully ask you to consider raising your score to reflect the improved quality of the manuscript.
>
> [1] UICopilot: Automating UI Synthesis via Hierarchical Code Generation from Webpage Designs. Gui. et al. WWW 2025
> [2] LaTCoder: Converting Webpage Design to Code with Layout-as-Thought Gui et al. KDD 2025.

---

### Note · Authors · 2025-12-25

I have read and agree with the venue's withdrawal policy on behalf of myself and my co-authors.